# The Bio-Aging of Biofilms on Behalf of Various Oral Status on Different Titanium Implant Materials

**DOI:** 10.3390/ijms24010332

**Published:** 2022-12-25

**Authors:** Min Liao, Yangyang Shi, Enni Chen, Yuke Shou, Dongyue Dai, Wenpan Xian, Biao Ren, Shimeng Xiao, Lei Cheng

**Affiliations:** 1State Key Laboratory of Oral Diseases, West China Hospital of Stomatology, National Clinical Research Centre for Oral Diseases, Sichuan University, Chengdu 610064, China; 2Department of Operative Dentistry and Endodontics, West China Hospital of Stomatology, Sichuan University, Chengdu 610041, China; 3Department of Periodontology, West China Hospital of Stomatology, Sichuan University, Chengdu 610041, China

**Keywords:** bio-aging, titanium, dental implant, biofilms, periodontitis

## Abstract

The properties of titanium implants are affected by bio-aging due to long-term exposure to the oral microenvironment. This study aimed to investigate probable changes in titanium plates after different biofilm bio-aging processes, representing various oral status. Titanium plates with different surface treatments were used, including polish, sandblasted with large grit and acid etched (SLA), microarc oxidation (MAO), and hydroxyapatite coating (HA). We established dual-species biofilms of *Staphylococcus aureus* (*S. aureus*)–*Candida albicans* (*C. albicans*) and saliva biofilms from the healthy and patients with stage III–IV periodontitis, respectively. After bio-aging with these biofilms for 30 days, the surface morphology, chemical composition, and water contact angles were measured. The adhesion of human gingival epithelial cells, human gingival fibroblasts, and three-species biofilms (*Streptococcus sanguis*, *Porphyromonas gingivalis,* and *Fusobacterium nucleatum*) were evaluated. The polished specimens showed no significant changes after bio-aging with these biofilms. The MAO- and SLA-treated samples showed mild corrosion after bio-aging with the salivary biofilms. The HA-coated specimens were the most vulnerable. Salivary biofilms, especially saliva from patients with periodontitis, exhibited a more distinct erosion on the HA-coating than the *S. aureus*–*C. albicans* dual-biofilms. The coating became thinner and even fell from the substrate. The surface became more hydrophilic and more prone to the adhesion of bacteria. The *S. aureus*–*C. albicans* dual-biofilms had a comparatively mild corrosion effect on these samples. The HA-coated samples showed more severe erosion after bio-aging with the salivary biofilms from patients with periodontitis compared to those of the healthy, which emphasized the importance of oral hygiene and periodontal health to implants in the long run.

## 1. Introduction

A dental implant, which often stays in the oral cavity for months, years or even life after implantation, is an effective way to repair missing teeth at present [1]. Titanium is widely used for implants due to its excellent biocompatibility, high mechanical strength, and good corrosion resistance [2]. However, the bio-activity and bone conductivity of titanium implants decrease gradually over time due to long-term exposure to the oral microenvironment, that is, biological aging [3]. Biological aging may lead to cracks on the surface of the materials and mismatched abutment connections, which will seriously affect the long-term survival rate of the prosthesis [4,5,6].

There are more than 700 species of microorganisms in the mouth, and oral bacteria begin to colonize the implant surface as soon as 30 min after implantation [7]. It was found that the long-term adhesion of pathogens to the surface of titanium implants will lead to the local corrosion of the material, a decrease in mechanical properties, the destruction of biological properties, and a decline in osseointegration ability [8,9]. As described in our previous studies, *Streptococcus mutans* and salivary microbe aging changed morphological features, mechanical properties, color, and the roughness of resin composites and glass ionomer cement [10]. Although titanium is relatively stable, long-term biological aging can degrade its properties [4]. Bio-aging with *Staphylococcus aureus* (*S. aureus*) biofilms could lead to the dissolution of the HA-coating and change hydrophilicity, surface roughness, and cytocompatibility of the HA-deposited titanium disks. [11] Additionally, our previous studies have also found that the aging treatment via saliva biofilms from healthy people significantly changed the properties of different surface-modified titanium plates, including surface morphology, the wettability, and the adhesion of cells and bacteria [3]. However, the composition and quantity of microbial communities vary among individuals under different health status. The initial formation and subsequent stability of peri-implant soft tissue structures and the corrosion of implants are unpredictable in different oral microenvironments. Therefore, it is essential to study the biological aging of modified abutment materials in different oral microenvironments [4,6,12].

A history of periodontitis and the presence of a periodontal pocket have been identified as risk factors for bacterial infection of the implant [13,14,15]. Meta-analysis showed that the risk of developing peri-implantitis in patients with periodontitis was 2.15 times higher than in those without periodontitis [14]. Additionally, the clinical symptoms and microbes of peri-implantitis are similar to that of periodontitis [16,17]. The number of bacterial species associated with periodontal disease etiology was significantly higher in patients with peri-implantitis than in the healthy [13,14].

*S. aureus* is one of the pathogenic bacteria involved in peri-implant inflammation, especially in suppurative sites. It has a specific affinity for titanium and is specifically detected in subgingival specimens around failed implants. *S. aureus* and *Candida albicans* (*C. albicans*) often colonize the surface of biomaterials or medical devices as a symbiont, causing severe suppurative infection at the corresponding sites. The simultaneous detection rate of *S. aureus* and *C. albicans* was higher in patients with peri-implantitis compared with that in the healthy [18,19,20].

These microorganisms around implants can create a favorable microenvironment for their own survival, which may destroy the homeostasis and impair the properties of the implants [21]. Therefore, we established microbial aging models of implant materials to simulate different oral microenvironments using the saliva of healthy people, saliva of people with periodontitis, and *S. aureus*–*C. albicans* dual-biofilms on behalf of the healthy, the patients with periodontitis, and patients with a severe suppurative infection in the cavity, respectively. Samples with different commercially available surface modifications were used, including polish, sandblasted with large grit and acid etched (SLA), microarc oxidation (MAO), and hydroxyapatite coating (HA). SLA increases surface roughness and provides mechanical interlocking between the bone tissue and implant, which is beneficial to the long-term mechanical stability of the bone–implant interface [22]. MAO produces a porous oxidized surface that can facilitate the direct contact and bonding between the bone and implant [23,24]. Another surface modification widely used is the HA-coating. Since its crystallographic, chemical, and mineralogical composition is similar to bone and teeth in humans, HA-coating can improve biocompatibility and bioactivity of the surface of titanium [25].

The aim of this study was to investigate the probable changes in properties of these samples with various surface modifications after the bio-aging treatment with different biofilms for 30 days so as to evaluate the effect of different oral microenvironments on the long-term performance of titanium implants.

## 2. Results

### 2.1. Surface Morphology

First, we photographed the specimens through a stereomicroscope. As shown in Figure 1a, the most obvious changes were observed in the HA-coated specimens. After bio-aging treatment by both types of saliva, the HA on the titanium plates was significantly missing and reduced, and the color of HA changed from gray to white, especially by the saliva of patients with periodontitis. Beyond that, there were no obvious changes in the titanium plates of other groups before and after aging.

Furthermore, the surface morphology of titanium plates was further clearly demonstrated by a scanning electron microscope (SEM) (Figure 1b). The polish-treated titanium plates in the three aging groups had uniform, regular straight grooves similar to the no treatment group that were produced due to the polish process. As for the SLA-treated samples, there were pits, cavities of various sizes, and sharp edges after the *S. aureus*–*C. albicans* dual-biofilm aging treatment, which is similar to the non-aging treatment group, while specimens after the bio-aging treatment with the saliva from healthy people/patients with periodontitis showed a surface with blurred cavities and relatively smooth edges. Similarly, the *S. aureus*–*C. albicans* dual-biofilm aging treatment did not change the surface morphology of the MAO-treated and HA-coated specimens; there were wire-like HA crystals in clear bundles on the HA-coated specimens and pores of uniform size on the MAO-treated specimens. After both saliva biofilm bio-aging treatments, the pores on the MAO-treated specimens appeared indistinct, and the HA crystals became blunt and unconsolidated. These results revealed that the saliva of heathy/periodontitis biofilm bio-aging treatment had a corrosive effect on titanium plates with all different surface modification treatments except the polish.

### 2.2. Surface Characteristic

The Raman spectrum was used to assess the chemical properties of the specimens. The SLA- and polish-treated specimens showed no distinctive peaks, so the results are not shown here. As shown in Figure 2a, there are four bands at 142, 236, 446, and 610 cm^−1^, respectively, in the MAO-treated specimens, which represents the rutile TiO_2_. These absorption peaks remain unchanged after the three models of bio-aging except that the peaks at 446 and 610 are decreased slightly after bio-aging via saliva biofilms from healthy people. As for the HA-coated specimens, there are three bands at 432, 584, and 961 cm^−1^, respectively, in the no treatment group, which represent the absorption peaks of PO_4_^3−^. After bio-aging via *S. aureus*–*C. albicans* dual-biofilms, the signal intensity of these peaks decreased, which might be due to a reduction in the material attached to the substrate of TiO_2_ [26]. Moreover, these peaks almost disappeared on the surface of samples bio-aged with both saliva biofilms except for a weak peak at 961 cm^−1^, which indicated the corrosion and destruction of the HA coating on the plates.

The wettability of the samples was measured by the water contact angle (Figure 2b). A smaller water contact angle means a better hydrophilicity [27]. Bio-aging with various biofilms did not change the water contact angle in the polish and MAO group. Bio-aging with saliva from the healthy people decreased the water contact angle in the SLA-treated samples. As for the HA-coated specimens, bio-aging with various biofilms decreased the water contact angle significantly, especially with the saliva from patients with periodontitis, which indicated that the saliva biofilms increased the hydrophilicity of HA-coated samples.

### 2.3. Adhesion and Proliferation of HGEs and HGFs

Human gingival epithelial cells (HGEs) and human gingival fibroblasts (HGFs) are the main cell types in the soft tissue around the implant. These cells attach to the surface of the implants and play an important role in host defense and the peri-implant soft tissue seal [28,29]. Here, we evaluated the adhesion and proliferation of these two kinds of cells on the surface of specimens. As shown in Figure 3a,b, the HGEs were more distinct and more likely to clump together on the polished specimens, while the morphology of the HGEs looked irregular on the HA-coated plates, probably because the cells hid among the HA crystals. No obvious difference was observed on the three bio-aging treatment groups compared to the no treatment group. The HGEs on the HA-coated specimens at 24 h looked relatively clear after bio-aging with the saliva of healthy people. This was due to the falling-off of the HA-coating on the surface of the titanium plates and the exposure of the substrate underneath, and some of the cells adhered to the surface of the titanium plate substrate where the HA loss occurred. Similarly, since the surface of the MAO and SLA groups were not smooth either, the cellular outline became ambiguous. After bio-aging via saliva from heathy people/patients with periodontitis, the pores and holes in the surface of titanium plates became smaller and the HGEs on these specimens appeared clearly visible. Likewise, the cellular morphology of HGFs looked ambiguous on the HA-coated sample (Figure 4a,b). There was no significant difference between polish-treated titanium plates before and after the three ways of bio-aging.

A Cell Counting Kit-8 (CCK-8) was used to measure the metabolic activities of cells at 24 h and 48 h (Figure 3c,d). After bio-aging with the saliva of healthy people, the metabolic activities of HGE on the polish-, MAO-, and SLA-treated plates increased at 24 h. After culturing for 48 h, the metabolic activities of HGEs in the aging groups showed no difference compared to the no aging group. As for the HGFs, the cells on polish-treated plates decreased after bio-aging via *S. aureus*–*C. albicans* dual-biofilms at 24 and 48 h. As for the MAO-treated specimens, the cells decreased after bio-aging via *S. aureus*–*C. albicans* dual-biofilms at 48 h. The cells on HA-coated specimens declined after bio-aging via the saliva of healthy people at 48 h compared to no bio-aged group (Figure 4c,d). Beyond that, there was no difference in the metabolic activities on the SLA-treated surface after bio-aging with the three kinds of biofilms.

### 2.4. Adhesion and Proliferation of Multispecies Biofilms

A multispecies biofilm of *Streptococcus sanguis* (*S. sanguis*), *Fusobacterium nucleatum* (*F. nucleatum*), and *Porphyromonas gingivalis* (*P. gingivalis*) was constructed. Then, the adhesion and metabolic activity of the biofilms on the surface of different titanium plates were measured using laser confocal microscopy and CCK-8. As shown in Figure 5a, green fluorescence represented live bacteria and red represented dead bacteria; these three bacteria could be seen on all specimens, and there were very few dead bacteria. On the HA-coated plates, bacteria were not distributed evenly and adhered in clumps; the adherent bacteria increased after bio-aging with both types of saliva and reduced after bio-aging with *S. aureus*–*C. albicans* dual-biofilms compared to the no bio-aging group. Similarly, on the SLA-treated plates, there were more adherent bacteria and they gathered into a mass after bio-aging with both types of saliva biofilms compared to no bio-aging. No obvious difference was observed on polish- and MAO-treated samples after bio-aging by all three ways.

Then, CCK-8 was used to measure the metabolic activities of the multispecies biofilms at 72 h (Figure 5b). There was no difference in the metabolic activities of the multispecies biofilms except for the HA group, in which the metabolic activities of the multispecies biofilms increased significantly after bio-aging with saliva from healthy people and saliva from patients with periodontitis.

## 3. Discussion

The long-term adhesion of pathogens to the surface of titanium implants can cause a disruption in homeostasis and a decline in the properties of biomaterials [3,10]. The objective of this study was to evaluate the influence of bio-aging with different biofilms on titanium plates. Samples with different surface modifications were used, including polish, SLA, MAO, and HA-coating. To simulate the biofilm bio-aging of titanium plates in different oral statuses, we collected saliva from ten individuals to represent the different oral status of the healthy and patients with periodontitis, respectively. Meanwhile, we cultivated *S. aureus*–*C. albicans* dual-biofilms to represent the bacterial community of the severe suppurative infection and evaluated the influence of bio-aging with these different biofilms after 30 days.

We used a stereomicroscope and SEM to observe the surface morphology. The polish-treated sample seemed the most stable, as it looked glossy and had some regular grooves due to the machining process, and the surface morphology did not change after the three ways of bio-aging (Figure 1). The MAO and SLA showed no changes in color, but SEM indicated that bio-aging with both types of saliva had a mild erosion effect on the samples, which manifested as the pores on the MAO-treated sample became smaller and the SLA surface became blunt. The HA-coated surface showed the most serious erosion after bio-aging with both types of saliva. The color of the HA-coated titanium plates changed from gray to white, and the HA coating became thinner and exhibited lost, which seemed worse when exposed to the saliva biofilms from patients with periodontitis. Moreover, bio-aging with the *S. aureus*–*C. albicans* dual-biofilms exhibited no obvious erosion on the surface of all samples.

The HA-coated samples in our study were made with plasma spray. Plasma spray is the only method that the FDA has approved commercially for biomedical coating on implants [30,31]. However, the major drawback of this technique is that the HA coating demonstrates poor adhesion and a relatively high thickness (>30 μm), which easily delaminates and falls from the substance [32]. On the other hand, the corrosion of the HA coating often happened in a high-alkaline/acid environment [3]. Generally, bacteria in saliva can easily adhere to the biomaterials in the cavity and accumulate to form biofilms once coming into contact. These microorganisms will create a favorable microenvironment for their own survival, which may destroy the homeostasis and impair the properties of the biomaterials [21]. The metabolites produced by the mixed microorganisms, such as lactic acid, might change the pH of the microenvironment, which could cause erosion and loss of the HA coating [33]. Our results showed that the HA-coated samples exhibited more serious corrosion after bio-aging with biofilms from patients with periodontitis than those of the healthy, which supported the hypothesis that the history of periodontitis is an important risk factor for peri-implantitis [34].

The Raman spectrum analysis was used here to detect the chemical structure of the samples. The SLA- and polish-treated specimens showed no distinctive peaks, so the results were not shown. The MAO-treated samples showed no significant changes via Raman spectrum analysis after bio-aging by these biofilms. As for the HA-coated titanium plates, there were three brands at 432, 584, and 961 cm^−1^, respectively, that represent the absorption peaks of PO_4_^3−^. The samples showed a relatively lower signal intensity after bio-aging with *S. aureus*–*C. albicans* dual-biofilms, which indicated a reduction in the adhesion of the coating materials and a mild effect of erosion that were not obvious enough to be observed by SEM or stereoscopic microscope. There was only a weak peak at 961 cm^−1^ for the samples in both saliva bio-aging groups, which indicated the loss of the HA coating on the substance and was consistent with the results from SEM and the stereoscopic microscope.

Wettability mainly depends on the roughness of biomaterials and indicates the hydrophilicity/hydrophobicity of materials. A smaller water contact angle means a better hydrophilicity [27]. The surface hydrophilicity of the titanium plate affects the adsorption of protein, cell adhesion, proliferation, and differentiation [35,36]. The HA-coated samples had the lowest water contact angle among the samples with four surface modifications. Bio-aging with salivary biofilms further reduced the water contact angle and improved the hydrophilicity of the HA-coated specimens, so we observed an increase in adhesion and proliferation of three-specie biofilms (*S. sanguis*, *P. gingivalis,* and *F. nucleatum*).

HGEs and HGFs are the main types of cells in the tissue around the implants. HGEs are located in the epithelial layer, and HGFs are present at the connective tissue below the epithelial layer. Their adhesion to the implants is important for establishing the tissue barrier peri-implant. These cells also play a key role in innate and antimicrobial host defense against infection [29]. Therefore, we investigated the adhesion and metabolism of HGEs and HGFs on the titanium plates. It was reported that epithelial cells and fibroblasts are liable to adhere firmly to smooth surfaces, while rough surfaces favor osteoblast proliferation and collagen synthesis [35]. Although these specimens were different in surface roughness in our study, all samples after bio-aging showed good affinity to these cells and were favorable for their adhesion and proliferation, except that the metabolic activity of HGFs decreased slightly after bio-aging with some biofilms (Figure 4c,d).

Bacteria can colonize an implant as soon as 30 min after insertion. The penetrating gingival site of an abutment made of titanium lacks the bacteriostatic effect and is prone to the attack of bacteria. Additionally, the lack of sharpey fibers and periodontal ligaments in implants could weaken the physical barrier of the oral mucosa and make it vulnerable to the invasion of bacteria [37], leading to implant-associated infections or even mobility or dislocation of implants [29]. To evaluate the affinity of the titanium plates for microorganisms after bio-aging, we evaluated the adhesion and proliferation of three-specie biofilms (*S. sanguis*, *P. gingivalis,* and *F. nucleatum*), which stand for colonizers in the early, middle, and late stages during biofilm formation, respectively [38]. As the pioneer in initial colonization, *S. sanguis* provides links for the coadhesion and coaggregation of the later species [39]. *P. gingivalis* is the predominant strain in the pathological or active part of peri-implantitis, and it participates in the beginning of inflammation and tissue destruction [40]. *F. nucleatum* is also an important pathogen in periodontitis and is specifically enriched in peri-implant diseases, which is proposed to be one of the peri-implantitis-related complexes that characterize peri-implantitis sites [41].

The rough surface of the implant is conducive to the integration of bone, as well as to microbial adhesion and colonization [3]. In our study, *S. sanguis* was the primary strain that adhered to all the samples among the three species (Figure 5). The metabolic activity of the three-species biofilms was highest on the HA-coated specimens (Figure 5). Bio-aging with both salivary biofilms further increased the metabolic activity, which might be due to the changes in roughness. When the surface roughness is greater than 0.2 μm, the bacterial colonization is promoted significantly [42]. The roughness of the SLA and HA are greater than 0.2 μm, and bio-aging of both types of saliva further increased the roughness. The higher roughness increased the surface area, which is beneficial for the adhesion and proliferation of bacteria.

In summary, our results showed that polished samples were most stable after the bio-aging of biofilms, and HA showed the most obvious corrosion after biofilm aging. The saliva biofilms had a worse effect on the titanium-based implant materials than *S. aureus*–*C. albicans* dual-biofilms, especially the saliva from patients with periodontitis. These results suggested that the periodontal health status is important for the long-term prognosis of implants. However, the difference in shape between implant and titanium plates we used in this study was a limitation of our experiment, which could be optimized in further studies. Additionally, the mechanism of implant bio-aging is not fully understood. There are some other factors jointly affecting the performance of implants in the complex oral microenvironment. Therefore, further research is needed to better simulate the microenvironment in vivo and evaluate its influence on implant materials comprehensively.

## 4. Materials and Methods

### 4.1. Subject Recruitment and Sampling

Ethical approval was obtained from the Ethics Committee of the West China Hospital of Stomatology Sichuan University (Chengdu, China) (license number WCHSIRB-D-2019-205). Saliva from healthy people was collected according to the following inclusion and exclusion criteria [3]. Healthy adults with natural dentition and no periodontal disease or active caries were included. Persons were excluded if they had used systemic antibiotics in the last three months or antibacterial mouthwash in the past 4 weeks. Saliva from people with periodontitis was collected with the following inclusion and exclusion criteria. Patients diagnosed with stage III–IV periodontitis according to the “2018 international classification of periodontal disease and implant disease: staging of periodontitis” were chosen [43,44,45]. Patients meeting one of the following criteria were excluded: edentulous patients; patients that had received systemic antibiotics in the last 3 months; patients that received systemic steroids or prophylactic antibiotics before the clinical examination; patients who took any medication known to affect periodontal conditions within 2 weeks; patients who had used antibacterial mouthwash in the past 4 weeks; patients who received any periodontal treatment in the past 6 months [46,47].

All donors mentioned above could not drink alcohol or brush their teeth within 24 h before sampling. They could not smoke or eat anything 3 h before the appointment. All sampling procedures were performed by the same dentist. Ten volunteers were selected and signed informed consent in each group. Equal amounts of saliva were collected from each volunteer, pooled, diluted twofold with 50% sterile glycerol, and stored at −80 °C.

### 4.2. Bacteria, Fungus and Culture Conditions

*S. aureus* (ATCC 25923), *C. albicans* (SC5314), *S. sanguis* (ATCC 10556), *F. nucleatum* (25586), and *P. gingivalis* (W83) were obtained from the State Key Laboratory of Oral Diseases (Sichuan University, Chengdu, China). *S. aureus* were incubated at 37 °C in Tryptone soya broth (TSB, Oxoid, Basingstoke, UK). *S. sanguis* were routinely incubated at 37 °C with 5% CO_2_ in brain heart infusion (BHI) broth (BD, Franklin Lakes, NJ, USA). *C. albicans* were cultured in YPD liquid medium (1% yeast extract, 2% peptone, 2% dextrose) at 35 °C. *F. nucleatum* and *P. gingivalis* were cultured in BHI broth supplemented with hemin (5 μg/mL), menadione (1 μg/mL), and 5% sheep deliberated blood in an anaerobic chamber at 37 °C.

### 4.3. Cell Culture

HGEs and HGFs were obtained from the State Key Laboratory of Oral Diseases (Sichuan University, Chengdu, China). The cells from two to sixth passage were used for all experiments. The cells were cultured in Dulbecco’s Modified Eagle Medium (DMEM, Gibco, Grand Island, New York, USA) supplemented with 10% FBS and 1% *v*/*v* penicillin/streptomycin at 37 °C in a humidified atmosphere with 5% CO_2_. The cells were detached with 0.25% trypsin (Hyclone, Waltham, MA, USA) and passaged after reaching 80–90% confluency.

### 4.4. Bio-Aging with Biofilms

Titanium plates with different surface modifications (polish, SLA, MAO, and HA) 6 mm in diameter and 1 mm in thickness were purchased from Chengdu Puchuan Biomaterials Co. Ltd. and sterilized with ethylene oxide. The titanium plates were divided into four groups, including no bio-aging, bio-aging with *S. aureus*–*C. albicans* dual-biofilms, saliva biofilm bio-aging from healthy people and saliva biofilm bio-aging from patients with periodontitis. Each group contained five plates with each surface treatment, respectively. For both saliva biofilm bio-aging groups, the titanium plates were placed in 24-well plates containing 980 μL SHI medium and a 20 μL saliva sample in each well. For the *S. aureus*–*C. albicans* dual-biofilm bio-aging group, the titanium plates were placed in a 24-well plate, *C. albicans* and *S. aureus* were inoculated on titanium plates in a final concentration of 10^6^ CFU/mL, respectively, with 1 mL RPMI1640 per well. All plates were incubated in an anaerobic incubator (90% N_2_, 5% CO_2_, 5% H_2_) at 37 °C, and medium was refreshed daily for 30 days. Then, all titanium plates were washed with an ultrasonic washer and sterilized with ethylene oxide for the following tests.

### 4.5. Surface Morphology Observation

The morphological characteristics of the titanium plate surface were observed using a stereomicroscope (Leica EZ4HD, Leica Microsystems AG, Heerbrugg, Switzerland) and photographed (*n* = 3). The scanning electron microscopy (SEM) (Quanta 200, FEI, Hillsboro, OR, USA) was performed to observe the surface morphology. All samples were sputter-coated with gold and observed with a magnification of 20,000 (*n* = 3).

### 4.6. Surface Chemical Changes

The surface chemical changes were evaluated by micro-Raman spectroscopy (DXR, Thermo Scientific, Waltham, MA, USA). Three sites selected randomly were analyzed. A 784 nm (infrared light) laser with 1 mW power was used for beam excitation. The spectra intensity was measured between 50–1600 cm^−1^ with 3 cm^−1^ spectral resolution. Spectra were considered identical when the Raman shifts of the peaks appeared at the same wavenumber and the relative intensities of the major peaks demonstrated less than 20% height difference.

### 4.7. Water Contact Angle

The hydrophilicity and hydrophobicity of the titanium plate surface were evaluated by the water contact angle. The droplets of 5 μL deionized water were dripped on the surface of the samples, and a picture was taken to record and calculate the static contact angle of titanium plates before and after bio-aging (Drop Shape Analyzer, KRUSS K100, Hamburg, Germany).

### 4.8. Adhesion and Proliferation of HGEs and HGFs on Titanium Plates

The titanium plates sterilized by ethylene oxide were placed in 48-well plates, and HGEs were seeded onto the titanium plates at a density of 1 × 10^4^ per well and cultured for 24 h and 48 h. Then, the titanium plates were transferred to a new 48-well plate; CCK-8 was diluted at a ratio of 1:10 with DMEM and added to the wells. The OD_450_ of the medium was measured after incubation at 37 °C in the dark for 1 h. Cells adhering to the titanium plates were stained by a live and dead cell staining reagent (Calcein-AM/PI, YOBIBIO, Shanghai, China), and observed with the laser scanning confocal microscope (FV1000, Olympus, Tokyo, Japan).The adhesion and proliferation of HGFs was measured in the same way.

### 4.9. Adhesion and Proliferation of Multispecies Biofilms

Multispecies biofilms were generated by using the method described by MSM Rigolin [48]. Briefly, the titanium plates sterilized by ethylene oxide were placed in 48-well plates. *F. nucleatum* (ATCC 25586), *P. gingivalis* (W83), and *S. sanguinis* (ATCC10556) in the logarithmic growth phase were inoculated in 100 μL BHI broth at a concentration of 10_9_ CFU/mL, and the total volume of each well was 300 μL. After incubation at 37 °C in an anaerobic incubator (90% N_2_, 5% CO_2_, 5% H_2_) for 72 h, the plates were washed twice with PBS to remove bacteria that were not adhered and transferred to new 48-well plates. Bacteria adhering to the titanium plates were stained by a live and dead cell staining reagent (Bestbio, Shanghai, China) and observed with the laser scanning confocal microscope. The metabolic activity of multispecies biofilms was measured by CCK-8 as described before [49].

### 4.10. Statistical Analysis

Data before and after bio-aging were analyzed using one-way ANOVA and Fisher’s LSD multiple comparison tests by SPSS statistical software, and *p* < 0.05 was considered to be significant.

## 5. Conclusions

The polish-, MAO- and SLA-treated specimens were relatively stable after bio-aging with biofilms. The HA-coated specimens were the most vulnerable to bio-aging. The *S. aureus*–*C. albicans* dual-biofilms exhibited a comparatively mild corrosive effect on these samples. The salivary biofilms caused serious corrosion on the HA-coated titanium plates, especially those from patients with periodontitis, which emphasizes the importance of oral hygiene and periodontal health to implants in the long run.

## Figures and Tables

**Figure 1 ijms-24-00332-f001:**
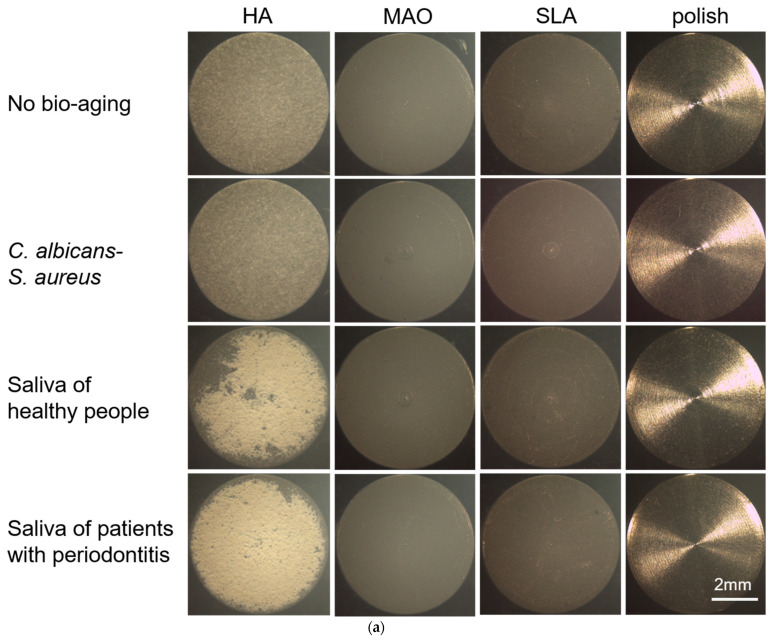
The surface morphology of samples as observed. (**a**) The gross morphology, such as color and morphology changes, were observed with a stereomicroscope; (**b**) The surface morphology is shown by SEM at a magnification of 20,000 (*n* = 3). Representative images are shown here.

**Figure 2 ijms-24-00332-f002:**
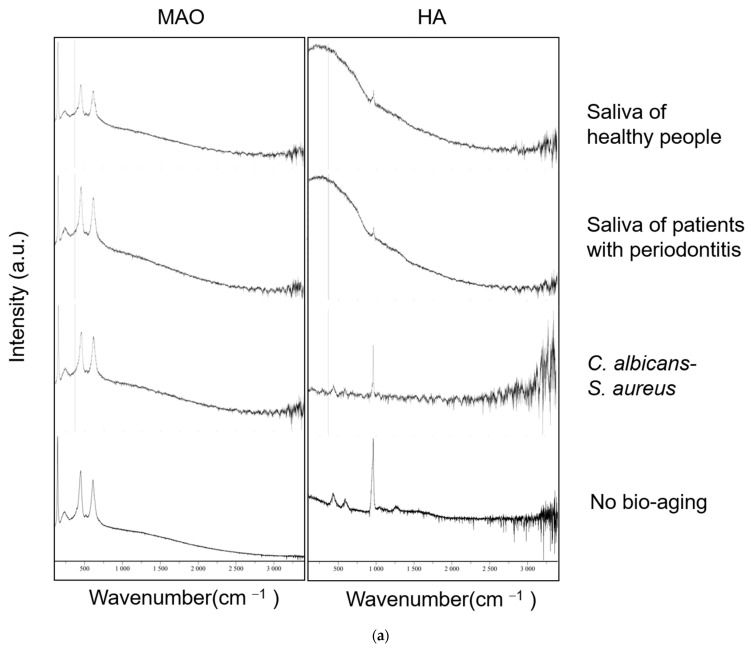
The surface characteristic of samples is tested. (**a**) Raman spectrum. The SLA- and polish-treated specimens showed no distinctive peaks so the results are not shown here; (**b**) water contact angle. The picture of the water droplet on the sample was placed above the bar. *n* = 5 (** *p* < 0.01).

**Figure 3 ijms-24-00332-f003:**
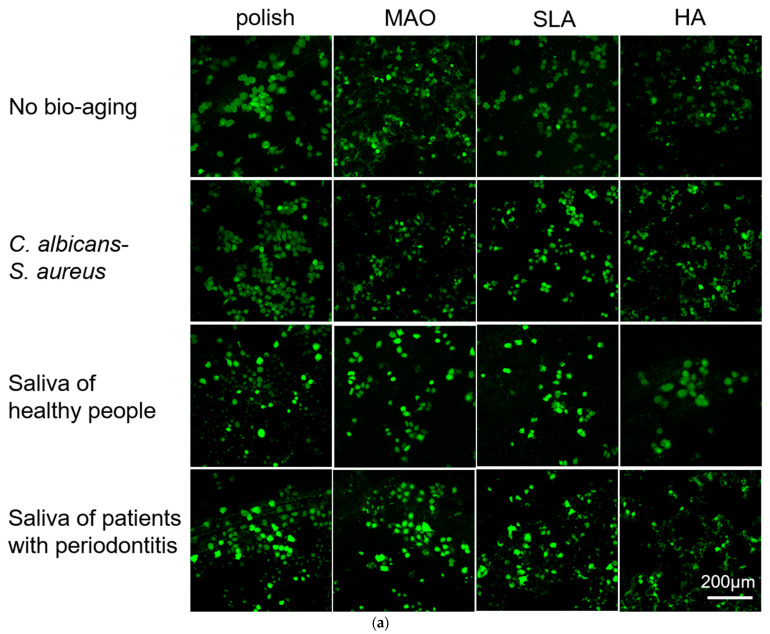
(**a**) and (**b**) The adhesion of HGEs was measured by laser confocal microscopy at 24 h and 48 h, respectively; (**c**) and (**d**) The metabolic activity of HGEs was measured by a CCK-8 test at 24 h and 48 h, respectively. *n* = 5 (* *p* < 0.05, ** *p* < 0.01).

**Figure 4 ijms-24-00332-f004:**
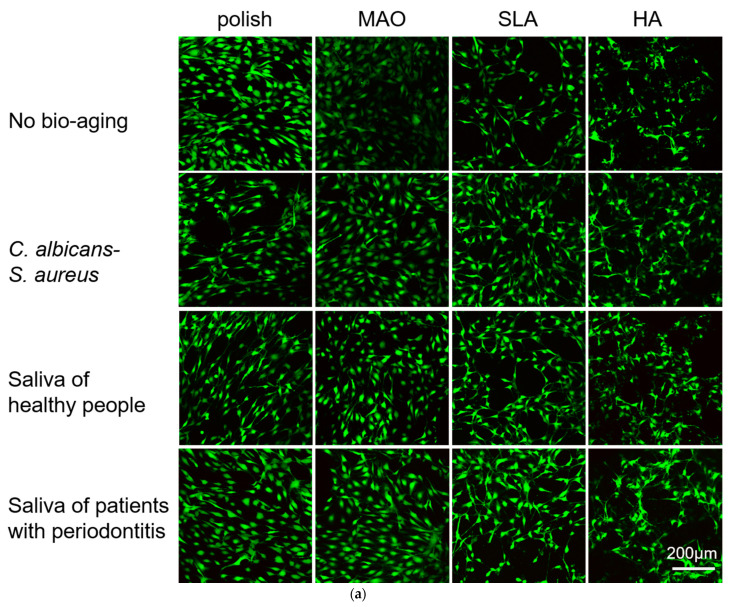
(**a**) and (**b**) The adhesion of HGFs was measured by laser confocal microscopy at 24 h and 48 h, respectively; (**c**) and (**d**) The metabolic activity of HGFs was measured by a CCK-8 test at 24 h and 48 h, respectively. *n* = 5 (* *p* < 0.05, ** *p* < 0.01).

**Figure 5 ijms-24-00332-f005:**
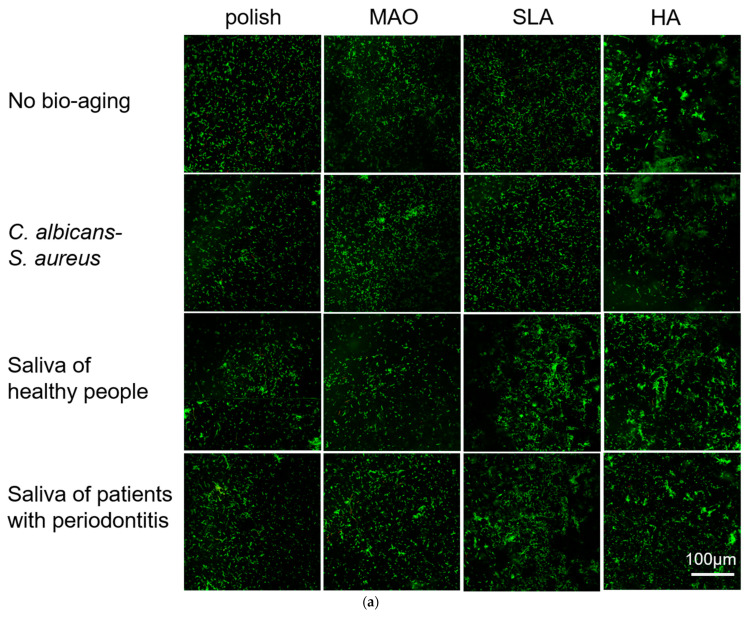
(**a**) The adhesion of multi-species biofilms (*S. sanguis*, *P. gingivalis,* and *F. nucleatum*) are measured by laser confocal microscopy at 72 h; (**b**) Metabolic activity is tested by CCK-8 test at 72 h. *n* = 5 (* *p* < 0.05, ** *p* < 0.01, *** *p* < 0.001).

## Data Availability

Not applicable.

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
