# Peer review of "The Bio-Aging of Biofilms on Behalf of Various Oral Status on Different Titanium Implant Materials"

_ijms, 2022, doi:10.3390/ijms24010332_

Round 1
Reviewer 1 Report
This manuscript, entitled " The bio-aging of biofilms on behalf of various oral status on different titanium implant materials" tried to evaluated the influence of bio-aging with different biofilms on titanium plates, and samples with four different surfaces modification were used. In addition, the authors put the titanium plates in four different oral status, saliva from healthy/patient and S. aureus-C. albicans dual-biofilms, for 30 days. The bio-aging effects of various biofilms on different titanium plates were well studied in this paper, which demonstrated that oral hygiene and periodontal health were important to implants in the long run.
Major comments
1. The figure 1b, the SEM of HA-coated samples were not clear compared to others. Can you provide SEM images with higher quality?
2. Line 377-378. In "Bio-aging with biofilms", why all plates were incubated in an anaerobic incubator (90% N2, 5% CO2, 5%H2) at 37°C? I think that S. aureus and C. albicans can growth better in air than in an anaerobic chamber.
3. To investigate the adhesion and proliferation of microbes, why did you choose the three-species biofilm model instead of the single-species biofilms or the saliva biofilms?
4. Since the water contact angel was photographed, please provide the picture to make the result more visual.
5. The introduction is redundant. Paragraph 1 and paragraph 2 can be merged and more refined.
6. The “2.2. surface characteristic”, the results and significance of water contact angle are not described in detail. Please describe these results and explain in detail.
7. When I skimmed through the whole manuscript quickly, I noticed that the format and size of some figures are not suitable. For example, compared with Figure 2a, the size of Figure 2b and Figure 5b is too small.
Minor comments
1. Line 79. Maybe "was" is wrong and should be replaced by "is".
2. Line 107. The "are" should be modified into "were".
3. It is suggested to add scale to figure 1a.
4. Line 356-357. For "hemin (5 mg/mL), menadione (1 mg/mL)", the final concentration of them should be 5 μg/mL hemin and 1 μg/mL menadione, respectively.
5. In figure 4c, the Y axis title “OD” should be changed into “OD450”.
6. Line 49-54 and line 72-74. The references are missing.
7. Line 360. Please describe the passage number of HGEs and HGFs used in this study.
Reviewer 2 Report
Dear Authors,
I have read the manuscript with interest and some questions raised. Please find my comments.
1. First of all, the writing sequence was wrong. Please check carefully, you should start with Introduction, Material and Methods, Results, Discussion and Conclusions.
2. Line 62: Besides, our previous studies also have found that the aging treatment of saliva biofilms from healthy people significantly changed the properties of different surface modified titanium plates, including surface morphology, the wettability, and adhesion of cells and bacteria [8].
If you can find more references involving bio-aging please add.
3. Please add the scale into Figure 3, 4 and 5 as same as you added in the Figure 1B (5 μm)
4. All bar graphs are too small, please increase the graph size. In addition, if you change the bar graph design to be color graphs, I think it will be more interesting.
5. Line 367: Titanium plates of different surface modification (Polish, SLA, MAO, and HA) were 6 mm in diameter and 1 mm in thickness and purchased from Chengdu Puchuan Bio-materials Co. Ltd. and sterilized with ethylene oxide.
Why you use titanium plates instead of the implant? What is the difference between using titanium plates comparing with the implant in this study? Please add the limitations of your study.
